# Expanded Access Programme for the use of tecovirimat for the treatment of monkeypox infection: A study protocol for an Expanded Access Programme

Josephine Bourner[1]*, Festus Devincy Redji Mbrenga[2], Christian Noël Malaka[2],
Jake Dunning[1], Amanda Rojek[1], Emmanuel Fandema[3], Peter Horby[1], Yap Boum, II[2],
Emmanuel Nakouné[2], Piero Olliaro[1]

1 Pandemic Sciences Institute, University of Oxford, Oxford, United Kingdom, 2 Institut Pasteur de Bangui,
Central African Republic, Bangui, Central African Republic, 3 Ministry of Health, Central African Republic,
Bangui, Central African Republic

☯ These authors contributed equally to this work.
* josephine.bourner@ndm.ox.ac.uk

## Abstract

### Background

Monkeypox is a viral zoonotic disease commonly reported in humans in parts of Central and
West Africa. This protocol is for an Expanded Access Programme (EAP) to be implemented
in the Central African Republic, where Clade I monkeypox virus diseases is primarily
responsible for most monkeypox infections. The objective of the programme is to provide
patients with confirmed monkeypox with access to tecovirimat, a novel antiviral targeting
orthopoxviruses, and collect data on clinical and virological outcomes of patients to inform
future research.

### Methods

The study will be conducted at participating hospitals in the Central African Republic. All
patients who provide informed consent to enrol in the programme will receive tecovirimat.
Patients will remain in hospital for the duration of treatment. Data on clinical signs and symp-
toms will be collected every day while the patient is hospitalised. Blood, throat and lesion
samples will be collected at baseline and then on days 4, 8, 14 and 28. Patient outcomes will
be assessed on Day 14 –end of treatment–and at Day 28. Adverse event and serious
adverse event data will be collected from the point of consent until Day 28.

### Discussion

This EAP is the first protocolised treatment programme in Clade I MPXV. The data gener-
ated under this protocol aims to describe the use of tecovirimat for Clade I disease in a mon-
keypox endemic region of Central Africa. It is hoped that this data can inform the definition of
outcome measures used in future research and contribute to the academic literature around
the use of tecovirimat for the treatment of monkeypox. The EAP also aims to bolster

doi.org/10.1371/journal.pone.0278957

CAMEROON

**Data Availability Statement:** No datasets were
generated or analysed during the current study. All

relevant data from this study will be made available upon study completion.

**Funding:** SIGA donated 100 treatments for use under the EAP. The EAP was financed with core funds of IPB, financial support to IPB from SIGA, the University of Oxford under the UK Foreign, Commonwealth and Development Office and Wellcome (215091/Z/18/Z) and the Bill & Melinda Gates Foundation (OPP1209135). Note: there are no grant award numbers for SIGA's donation to of tecovirimat to IPB, core funds of IPB, or the financial support of SIGA to IPB. The funders had and will not have a role in study design, data collection and analysis, decision to publish, or preparation of the manuscript.

**Competing interests:** The authors have declared that no competing interests exist.

**Abbreviations:** CAR, Central African Republic; Co-I, Co-Investigator; CRF, Case Report Form; D, Programme day; IB, Investigator's Brochure; IMP, Investigational Medicinal Product; NTD, Neglected Tropical Disease; PI, Principal Investigator; PIS, Patient Information Sheet; RCT, Randomised Controlled Trial.

research capacity in the region in order for robust randomised controlled trials to take place for monkeypox and other diseases.

## Trial registration

**{2a & 2b}**: ISRCTN43307947.

---

# Background

## Background and rationale {6a}

Monkeypox is a viral zoonotic disease commonly reported in humans in parts of Central and West Africa, including the Central African Republic (CAR), the Democratic Republic of Congo (DRC), Republic of Congo, Nigeria, Sierra Leone and Liberia [1]. There are two genetic clades of the monkeypox virus–Clade I has typically circulated in Central Africa and is associated with higher mortality [2], and Clade II in West Africa–as well as a sub-clade, Clade IIb, which was identified as being responsible for the multi-country outbreak of monkeypox in typically non-endemic countries starting in May 2022 [3].

This protocol is for an Expanded Access Programme (EAP) to be implemented in CAR, where Clade I MPXV is primarily responsible for most monkeypox infections (Institut Pasteur de Bangui [Unpublished]). There have been approximately 100 cases of monkeypox reported in CAR since 2000 –primarily in South-Western (Lobaye, Mambere-Kadei, Sanga-Mbaere), and South-Eastern (Mbomou) prefectures [4]. Reports of MPXV in humans in CAR have increased in recent years partly due to improvements in surveillance, but also as a result of the cessation of the smallpox vaccination programme–leading to reduced cross-protection against orthopox viruses–civil unrest and rampant deforestation, which is increasing contact between wildlife and humans [5,6]. In CAR, monkeypox has been reported in patients who have hunted wildlife [5] and is associated with harvesting caterpillars–a seasonal activity which sees individuals and families spend prolonged periods of time in the rainforests, often coming in to contact with bushmeat.

Patients who are diagnosed with monkeypox have limited treatment options. Until recently, supportive care to alleviate monkeypox symptoms–such as lesions, fever, headache, back and muscle aches, and lymphadenopathy–was the only treatment available [7]. However, tecovirimat–a novel antiviral targeting orthopox viruses by inhibiting the formation of the virus envelope, thus preventing its exit from infected cells–has recently been approved in several countries under exceptional circumstances in response to the 2022 multi-country monkeypox outbreak [8]. At the time this EAP was started, tecovirimat was only approved by the United States Food and Drug Administration (FDA) for smallpox under the so-called "animal rule" [9]. In all cases, despite receiving regulatory approval for the treatment of monkeypox, there is limited efficacy and safety data for the use of tecovirimat for monkeypox. Previous clinical data supporting tecovirimat are essentially on safety and tolerability from studies in healthy volunteers; there are only a handful of reports on tecovirimat used to treat patients with confirmed monkeypox infection, none of which are from randomised controlled clinical trials (RCTs) [10–13].

As with many other outbreak-prone infectious diseases, initiating an RCT for monkeypox poses several challenges. Firstly, monkeypox is reported sporadically over widespread geographical areas in rural parts of CAR, and other endemic regions, with limited healthcare infrastructure and poor transport links, making the implementation of a clinical trial challenging. Monkeypox is also reported in relatively small numbers of patients [4] and, while the case

fatality rate in Clade I MPXV is around 10% (Institut Pasteur de Bangui [Unpublished]), mortality is too low for it to be used as the primary endpoint in any clinical trial–the overall required sample size would be unfeasibly high for an effect to be detected between treatment groups. Alternative endpoints therefore need to be identified, which is complicated by the diversity of symptoms experienced by patients with monkeypox and lack of clinical and virological data on patient outcomes [14].

While conducting an RCT may not be an option now in CAR, conducting an EAP of tecovirimat to treatment monkeypox in CAR therefore has several benefits: the EAP will provide patients with access to a potentially life-saving treatment that would not otherwise be available; it will allow the collection of data on signs and symptoms, and clinical and virological outcomes of patients under a standardised protocol; and finally build research capacity to conduct future clinical research studies, including RCTs using an Investigational Medicinal Product (IMP).

## Methods

### Objectives {7}

The primary objective of this EAP is to provide access to tecovirimat to patients with monkeypox who have no comparable or satisfactory therapeutic alternative and cannot participate in a clinical trial.

The secondary objective of this study is to describe the clinical, virological and safety outcomes of patients with monkeypox virus disease.

### Methods/Design {8}

This protocol has been written according to the SPIRIT guideline (S1 Text).

This is an Expanded Access Programme (EAP) providing treatment for monkeypox virus disease to patients with PCR confirmed monkeypox in CAR.

All patients who sign the informed consent form (S2 Text) and enrol in the programme will receive tecovirimat–this is not a randomised study.

Data are collected on signs and symptoms and treatment administration every day while patients are hospitalised and receiving treatment. Data on patient outcomes are collected on D14 and D28. Adverse events (AEs) and Serious Adverse Events (SAEs) are collected from the point of consent until D28.

Blood, throat, and lesion samples are collected whenever possible at enrolment, D4, D8, D14 and D28. An additional blood, throat and lesion sample will be collected at D21 if patients test positive on any sample at D14.

While the primary objective of this EAP is to provide patients with access to treatment, the data collected in this study will be used to evaluate the outcomes of patients who receive tecovirimat. These data will contribute to the academic literature about the use of tecovirimat for the treatment of Clade I monkeypox.

This protocol also aims to build clinical research capacity in CAR.

### Ethics approval and consent to participate {24}

Ethical approval was obtained from the Oxford Tropical Research Ethics Committee (OxTREC) (ref: 1–20) and the University of Bangui Ethics Committee before the first patient was enrolled in the programme. All patients provide written informed consent before enrolment.

## Trial registration {2a & 2b}

This EAP is registered on the ISRCTN registry with the following identifier:
ISRCTN43307947.

## Programme setting {9}

This EAP is taking place in participating hospitals in the Central African Republic.

## Eligibility criteria {10}

**Inclusion criteria.** This programme is enrolling male and female patients who meet the
following criteria:

- Confirmation of monkeypox virus disease by PCR, or patients who are being managed as a
presumptive case pending laboratory confirmation*

- Weight ≥13 kg

- Willing to provide informed consent to enrol in the programme

*A blood, pharyngeal, and lesion swab are taken for confirmation by qPCR positivity for
monkeypox virus DNA, but enrolment will not await laboratory confirmation if there is high
clinical suspicion of disease. A patient who is a presumptive case and subsequently tests nega-
tive for MPXV will exit the study. A positive PCR result on any of the samples constitutes evi-
dence of MPXV infection.

**Exclusion criteria.** Patients will be excluded from the study if they meet any of the below
criteria:

- Co-administration of repaglinide

- Co-administration of midazolam

- Galactose intolerance, total lactase deficiency or glucose-galactose malabsorption

## Who will take informed consent? {26a}

All patients who meet the eligibility criteria are informed about the programme by the Princi-
pal Investigator (PI) or Co-Investigator (Co-I) in CAR. Interested patients are given the
Patient Information Sheet (PIS) (S2 Text), which is available in either French or Sango.
Patients will be given sufficient time to read the (PIS), consider their participation and discuss
the programme with the PI or Co-I.

The PI and Co-I will obtain written informed consent from all patients (or their legal repre-
sentatives) enrolled in the programme before commencing any study-specific procedures.

Adult patients are defined as those being 18 years old and above.

If an adult patient does not have capacity to provide informed consent, written advice will
be sought by the PI or Co-I from a close relative who attends the health centre with the patient
(a personal consultee) or an appropriate independent individual (nominated consultee). If the
patient later regains capacity, they will be provided with the PIS and asked to consider whether
they wish to continue participating in the programme.

For all children aged up to 17 years, key elements of the programme will be explained by
the PI or Co-I in a manner that is appropriate for the child's age and maturity. Verbal assent
will be sought from the child and written informed consent will be sought from a parent or
legal representative.

**Table 1. Dosage for adult patients weighing ≥40kg.**

| Body weight | Dosage | Number of capsules |
|---|---|---|
| 40kg to 120kg | 600mg twice daily | 3 capsules twice daily |
| ≥120kg | 600mg three times daily | 3 capsules three times daily |

## Additional consent provisions for collection and use of participant data and biological specimens in ancillary studies {26b}

Not applicable.

## Interventions

### Explanation for the choice of comparators {6b}

This is a non-comparative EAP. All patients enrolled in the study will receive tecovirimat according to the Investigator's Brochure (IB).

### Intervention description {11a}

Patients will be treated as inpatients at participating centres. Tecovirimat is administered by the PI or Co-I in CAR based at the participating district hospital according to the IB.

Tecovirimat should be taken within 30 minutes after a meal ideally of moderate or high fat. Treatment is given for 14 days following the dosing regimen described in Tables 1 and 2.

### Criteria for discontinuing or modifying allocated interventions {11b}

No modifications to treatment administration are permitted under this protocol.

Discontinuation of treatment will only take place on the request of the patient or at the discretion of the PI or Co-I (e.g. as a result of an intolerable adverse event). Discontinuation of treatment will be considered withdrawal from the study and the reason for withdrawal will be recorded on the Case Report Form (CRF).

### Strategies to improve adherence to interventions {11c}

Treatment is administered by the PI or Co-I to ensure adherence.

Drug dispensing is recorded on a drug accountability log on which any unused capsules are also recorded.

### Relevant concomitant care permitted or prohibited during the programme {11d}

Concomitant care is given according to local guidelines.

Repaglinide and midazolam are prohibited while the patient is receiving tecovirimat.

**Table 2. Dosage for adults weighing <40kg and children.**

| Body weight | Dosage | Number of capsules |
|---|---|---|
| 13kg to 24kg | 200mg twice daily | 1 capsule twice daily |
| 25kg to 39kg | 400mg twice daily | 2 capsules twice daily |
| ≥40kg | 600mg twice daily | 3 capsules twice daily |

**Table 3. Outcome measures.**

| Assessment of... | Outcome measure |
|---|---|
| Clinical outcomes | Time to lesion(s) resolution, defined by:<br>From a start point of date that treatment started.<br>Until an endpoint of up to 14 days since treatment start.<br>Where lesion resolution is the first day on which all lesions are resorbed, scabbed or desquamated and mucosal ulcers healed, and in the absence of any serious complications |
| | Clinical status defined by:<br>Clinical status on day 14 and day 28 according to an ordinal scale assessed by a physician. The ordinal scale is<br>a) no new or active lesions and no serious complications,<br>b) active lesions and no serious complications,<br>c) serious complication of monkeypox,<br>d) death. |
| Virological outcomes | Virological status defined by<br>Change from baseline in Monkeypox virus DNA levels in throat swabs on days 4, 8, 14 and 28.<br>Change from baseline in Monkeypox virus DNA levels in blood on days 4, 8, 14 and 28.<br>Presence of Monkeypox virus DNA in lesion swabs on days 4, 8, 14 and 28. |
| Safety outcomes | Number and type of Serious Adverse Events (SAEs), Suspected Adverse Reactions (SARS) and Suspected Unexpected Serious Adverse Reactions, (SUSARs) within 28 days of enrolment. |

## Provisions for post-programme care {30}

After the patient's final programme visit at D28, responsibility for clinical care is returned to the local healthcare system.

If a patient suffers harm as a result of their involvement in the programme, the Sponsor (the University of Oxford) has specialist insurance in place: Newline Underwriting Management Ltd, at Lloyd's of London.

## Outcomes {12}

Treatment effects will be evaluated based on clinical, virological and safety outcomes described in Table 3.

## Participant timeline {13}

The participant timeline is described in Fig 1.

## Sample size {14}

This is an EAP with a primary objective of providing treatment to patients with monkeypox. A formal sample size has therefore not been calculated. Ethical approval has been obtained however for treating 100 patients.

## Recruitment {15}

An active monkeypox surveillance programme has been put in place to detect and report cases of monkeypox in the target Prefectures. The PI and Co-PI have conducted a community engagement programme to train all health personnel working at health facilities and strengthen capacity in the clinical diagnosis of monkeypox. Essential equipment has also been provided to health facilities across the Prefecture to assist with the diagnosis of monkeypox.

## Assignment of interventions: Allocation

## Sequence generation {16a}

This is not a clinical trial and patients are not randomised to treatment. All enrolled patients receive tecovirimat.

| | Enrolment | Allocation | Post-allocation | | | | | | | | | | | | | | Follow-up | |
|---|---|---|---|---|---|---|---|---|---|---|---|---|---|---|---|---|---|---|
| **TIMEPOINT** | $-t_1$ | 0 | $t_1$ | $t_2$ | $t_3$ | $t_4$ | $t_5$ | $t_6$ | $t_7$ | $t_8$ | $t_9$ | $t_{10}$ | $t_{11}$ | $t_{12}$ | $t_{13}$ | $t_{14}$ | $t_{21}{}^a$ | $t_{28}$ |
| **ENROLMENT:** | | | | | | | | | | | | | | | | | | |
| **Eligibility screen** | X | | | | | | | | | | | | | | | | | |
| **Informed consent** | X | | | | | | | | | | | | | | | | | |
| **Comorbidities** | X | | | | | | | | | | | | | | | | | |
| **Pregnancy test[b]** | X | | | | | | | | | | | | | | | | | |
| **HIV test[b]** | X | | | | | | | | | | | | | | | | | |
| **Malaria RDT** | X | | | | | | | | | | | | | | | | | |
| **Allocation** | | X | | | | | | | | | | | | | | | | |
| **INTERVENTIONS:** | | | | | | | | | | | | | | | | | | |
| **Tecovirimat administration** | | | X | X | X | X | X | X | X | X | X | X | X | X | X | X | | |
| **ASSESSMENTS:** | | | | | | | | | | | | | | | | | | |
| **Blood sample for MPXV PCR test[c]** | X | | | | | X | | | | X | | | | | | X | X | X |
| **Throat sample for MPXV PCR test[c]** | X | | | | | X | | | | X | | | | | | X | X | X |
| **Lesion sample for MPXV PCR test[c,d]** | X | | | | | X | | | | X | | | | | | X | X | X |
| **Signs and symptoms** | X | | X | X | X | X | X | X | X | X | X | X | X | X | X | X | X | X |
| **Clinical outcome** | | | | | | | | | | | | | | | | X | X | X |

**Fig 1. Schedule of enrolment, interventions and assessment.** [a] The D21 visit will only be conducted for patients who test positive for monkeypox on D14 and remain hospitalised. [b] These tests will be voluntary. [c] Conducted +/- 24h of specified visit. [d] Two swabs taken per timepoint. Please see below for further details.

### Concealment mechanism {16b}

This is not a clinical trial and patients are not randomised to treatment. All enrolled patients receive tecovirimat.

### Implementation {16c}

This is not a clinical trial and patients are not randomised to treatment. All enrolled patients receive tecovirimat.

## Assignment of interventions: Blinding

### Who will be blinded {17a}

No one involved in the EAP is blinded to the intervention.

### Procedure for unblinding if needed {17b}

As the intervention is not blinded, there will no unblinding procedure.

## Data collection and management

### Plans for assessment and collection of outcomes {18a}

All data collected under the EAP are recorded on a paper CRF (**S1 Table**) by the PI or Co-PI and entered on to a RedCap database within 24 hours by the data manager who is also based at the district-level hospital.

All members of staff involved in the EAP have undergone training on the assessment of patient outcomes, including the differentiation between lesion types and reporting SAEs and SUSARs.

All blood, throat and lesion samples collected under this protocol are sent to the national reference laboratory at Institut Pasteur de Bangui for testing by G2R-G PCR assay–for the detection of MPXV (any clade)–and C3L PCR assay–for the detection of Clade I MPXV. The MPXV positive or negative result for each sample type are recorded on to the paper CRF and transferred on to the RedCap database.

### Plans to promote participant retention and complete follow-up {18b}

Patient outcomes in this programme are assessed at Days 14 and 28.

Patients who are enrolled in this programme will remain in hospital for a minimum of 14 days until the completion of treatment or, if patients are still positive for MPXV at Day 14, until they receive a negative PCR result and are clinically well to be discharged. All treatment will therefore be administered under the supervision of the PI or Co-PI.

Following discharge, patients will attend a final visit at Day 28.

### Data management {19}

Data are recorded on a paper CRF and then entered on to a RedCap database by the data manager based at the district-level hospital with other members of the study team.

The RedCap database contains data quality rules to prevent entry of erroneous data.

Centralised data monitoring is also implemented by the Sponsor to ensure completeness and accuracy of data relating to the primary and secondary outcome measures and patient safety.

The RedCap database will be held at the University of Oxford for a minimum of 25 years.

## Confidentiality {27}

All data and samples are labelled only with the patient's anonymous subject ID code.

The consent form and enrolment log will remain at the site of enrolment during the study and then transferred to the Institut Pasteur de Bangui to be kept securely for a minimum of 3 years. This information will not be shared with any individual or organisation outside the EAP team in CAR.

Electronic data will be held only on the RedCap database accessible to specified, individual users with unique logins who are involved in data collection, monitoring and analysis under this protocol.

## Plans for collection, laboratory evaluation and storage of biological specimens for genetic or molecular analysis in this trial/future use {33}

There are no plans to conduct genetic or molecular analysis on biological samples collected from patients enrolled in this EAP.

Patients can however consent to the use of their biological samples for future research. Consent to future use of biological samples is optional. Future research undertaken on these samples will undergo relevant reviews by national ethics committees and/or institutional review boards before any further analyses are undertaken.

# Statistical methods

## Statistical methods for analysing outcomes {20a}

Clinical and virological outcomes will be assessed at D14 and D28 for all patients and then at D21 for patients who are still positive at D14. Due to the non-comparative nature of the study and small number of patients anticipated to be enrolled in the study, only descriptive analyses of clinical and virological outcomes will be presented in the final analysis; no inferential statistical testing will be performed to evaluate changes in outcomes.

To describe the cohort, patient demographics will be described in terms of the ratio of males to females in the cohort, and the median age and range in years. The number and percentage of patients with comorbidities at inclusion, including malaria and human immunodeficiency virus, infection will be presented, along with the median interval in days from symptom onset to initiation of treatment.

Signs and symptoms will be reported in terms of the number and percentage of patients presenting with the symptom, which are assessed at each visit and will be reported at baseline and post-baseline timepoint. Signs and symptoms will be reported for all patients both from a pre-defined list included in the CRF and any additional AEs that may have arisen from the point of consent until D28. The signs and symptom reporting will also include the number and percentage of patients with lesions–subcategorised in to active lesions, crusts and resolved lesions–their distribution and number of lesions observed.

Treatment adherence will be reported in terms of the number and percentage of patients who completed a full course of treatment. Non-adherence will be reported in a supplementary narrative explanation.

Patient outcomes will be reported as the number and percentage of participants who meet the clinical criteria described in Table 3 at D14 and D28, along with the number and percentage of participants who tested positive for mpox virus on D4, D8, D14, D21 and D28.

Narrative summaries of SAEs will be presented alongside a tabulated summary of the number and percentage of patients who experience each SAE, for which event names will be MedDRA coded. For each SAE the grade and outcome will be reported.

Full details of the planned analyses can be found in the Statistical Analysis Plan.

### Interim analysis {21b}

Descriptive analyses of the cohort may be published before the end of the EAP in the interest of communicating information about patient outcomes and the use of tecovirimat for monkeypox which may be pertinent to other research efforts and public health. One such analysis of the first 14 patients enrolled in the EAP was published in September 2022 [15].

### Methods for additional analyses {20b}

All planned analyses on the final dataset will be specified in the Statistical Analysis Plan (SAP).

### Methods in analysis to handle protocol non-adherence and any statistical methods to handle missing data {20c}

The reporting of the study results will consist of descriptive statistics, covering patient demographics, signs and symptoms, clinical outcomes, virological outcomes, and treatment data. Missing data will be clearly reported through the use of denominators or narrative explanations of protocol non-adherence.

### Plans to give access to the full protocol, participant level-data and statistical code {31c}

The full protocol, anonymised participant level-data and statistical code will be either published alongside the study results as supplementary files or available on reasonable request to the programme coordination team.

## Oversight and monitoring

### Composition of the coordinating centre and trial steering committee {5d}

The programme coordinating centre is based at the Institut Pasteur de Bangui with support from the University of Oxford, the study Sponsor. A Programme Management Group oversees day-to-day operational issues relating to the conduct of the programme. The group is formed of members of the coordinating centre and Sponsor and meets at least every two weeks while patients are enrolled during the monkeypox transmission season (September to March). The Programme Management Group is formed of the PIs, co-investigators, laboratory manager, clinical trials manager, and data manager.

A Programme Steering Committee provides overall oversight of the programme and convenes at the beginning and end of the transmission season. The committee is formed of the PIs, co-investigators, clinical trials manager and independent experts in monkeypox.

A medical monitor has been appointed to oversee safety-related aspects of the programme.

### Composition of the data monitoring committee, its role and reporting structure {21a}

As this is not a clinical trial, a data monitoring committee has not been convened. A medical monitor has instead been appointed to oversee issues related to patient safety.

The responsibilities of the medical monitor include, but are no limited to, commenting on any safety considerations related to the protocol, reviewing all Serious Adverse Event (SAE) reports and provide a causality assessment of the relationship between the event and tecovirimat, to periodically review a line listing of all AEs reported, to review all major protocol deviations, and provide advice on their management.

## Adverse event reporting and harms {22}

Details of Adverse Events (AEs) and Serious Adverse Events (SAEs) are collected from the point of consent until the Day 28 visit.

AEs and SAEs are defined and reported according to the WHO Collaborative Centre definitions incorporated in to the European Medicines Agency Guideline for Good Clinical Practice E6 [16].

AEs are recorded directly on the standardised paper CRF and transferred on to the RedCap database by the data manager. The data collected for all AEs includes the event name, event grade, start and end dates and outcome.

SAEs require expedited reporting to the Sponsor. SAEs are defined as any untoward medical occurrence that:

- results in death

- is life-threatening

- requires inpatient hospitalisation or prolongation of existing hospitalisation

- results in persistent or significant disability/incapacity

- consists of a congenital anomaly or birth defect.

Other 'important medical events' may also be considered a serious adverse event when, based upon appropriate medical judgement, the event may jeopardise the participant and may require medical or surgical intervention to prevent one of the outcomes listed above.

All SAEs should be reported to the Sponsor and PI on the standardised SAE report form within 24 hours of the site-based programme team becoming aware of the event.

The treating clinician (the PI or Co-I) will provide a causality assessment of the relationship between the event and tecovirimat. The relationship will be evaluated as either: 1) definitely related; 2) probably related; 3) possibly related; 4) unlikely to be related; 5) not related. The Medical Monitor will provide a second causality assessment using the same scale. Events will be considered as "related" to tecovirimat if either causality assessment is returned as "definitely", "probably", or "possibly" related. These events will be considered as Serious Adverse Reactions (SARs). Events that are considered to be "unlikely" or "not related" will remain as SAEs.

The Medical Monitor will also assess the expectedness of the event against the Investigator's Brochure (IB). Related events that are unexpected will be considered SUSARs.

All SAEs must be reported to the ethics committee of the University of Bangui within 72h of the programme team becoming aware of the event.

All SARs must be reported to the Oxford Tropical Medicine Ethics Committee (OxTREC) within 7 working days.

All AEs and SAEs event terms will be coded according to the Medical Dictionary for Regulatory Activities (MedDRA) terminology.

## Frequency and plans for auditing trial conduct {23}

No audit of the conduct of this programme is planned.

## Plans for communicating important protocol amendments to relevant parties (e.g. trial participants, ethical committees) {25}

All amendments to the protocol will be reviewed and approved by the responsible ethics committees before implementation.

### Dissemination plans {31a}

The results of the programme will be submitted to an open-access, peer-reviewed academic journal for publication along with the full dataset. The results are also shared regularly with the Ministry of Health in CAR. Communication of the results to local communities and stakeholders in CAR will be the responsibility of the Ministry of Health in CAR and the Institut Pasteur de Bangui.

### Authorship guidelines {31b}

Publications arising from this programme will be drafted by Steering Committee members and approved by the PI before submission. Authorship will be agreed among the Steering Committee and defined according to ICMJE guidelines.

## Discussion

In this protocol we describe an Expanded Access Programme for the use of tecovirimat to treat human monkeypox virus infection in the Central African Republic.

The Lobaye Prefecture was selected as the region in which the study would be conducted in the first place for several reasons: cases of monkeypox have historically frequently been reported in this area; there are good transport links to Bangui, meaning that samples can be transported easily between the site and the reference laboratory and Institut Pasteur de Bangui with limited delay; and the security risk in the prefecture is low. However, the catchment area will be adapted to the epidemiology of the disease in the country.

Due to limited healthcare infrastructure, one treatment site has been initially selected for the EAP. The site is located in a central town with good transport connections to other villages within the region. However, patient access to the site can be challenging. For this reason, the EAP provides transport for suspected cases of monkeypox identified in remote villages through the surveillance programme who would not otherwise not be able to reach the treatment centre.

A team of clinicians and programme support staff have also been employed locally to work on the EAP. This team has been deployed to the treatment centre and will oversee all aspects of the programme conduct, including patient treatment and data collection. This support and oversight is critical to ensure that the research burden is minimised for the health facility staff, patient safety is upheld and data quality is maintained.

In order to promote comparability of data generated between research studies, the outcome measures defined this protocol were selected in line with those of other ongoing studies [17,18]. Equally, the CRF was aligned with that of other studies to ensure that data is collected consistently and can be reported in a comparable manner. However, minor modifications have been made to account for the context in which the EAP is being conducted.

Overall, this EAP is the first protocolised treatment programme in Clade I MPXV. The data generated under this protocol aims to describe the use of tecovirimat for Clade I disease in a monkeypox endemic region of Central Africa. It is hoped that this data can inform the definition of outcome measures used in future research and contribute to the academic literature around the use of tecovirimat for the treatment of monkeypox. The EAP also aims to bolster research capacity in the region in order for robust randomised controlled trials to take place for monkeypox and other diseases.

## Supporting information

**S1 Table. Case report form.**
(DOC)

**S1 Text. SPIRIT checklist.**
(DOC)

**S2 Text. Model consent form.**
(DOCX)

## Author Contributions

**Conceptualization:** Josephine Bourner, Peter Horby, Emmanuel Nakouné, Piero Olliaro.

**Funding acquisition:** Peter Horby.

**Investigation:** Festus Devincy Redji Mbrenga, Christian Noël Malaka, Jake Dunning, Emmanuel Fandema, Yap Boum, II, Emmanuel Nakouné.

**Methodology:** Josephine Bourner, Jake Dunning, Amanda Rojek, Piero Olliaro.

**Project administration:** Josephine Bourner, Festus Devincy Redji Mbrenga.

**Resources:** Josephine Bourner, Festus Devincy Redji Mbrenga, Christian Noël Malaka, Yap Boum, II, Emmanuel Nakouné, Piero Olliaro.

**Supervision:** Emmanuel Nakouné, Piero Olliaro.

**Writing – original draft:** Josephine Bourner.

**Writing – review & editing:** Josephine Bourner, Festus Devincy Redji Mbrenga, Jake Dunning, Amanda Rojek, Peter Horby, Emmanuel Nakouné, Piero Olliaro.

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
