## [Decision Letter · Decision Letter 0]

14 Feb 2023

PONE-D-22-32294

Expanded Access Programme for the use of tecovirimat for the treatment of monkeypox infection: a study protocol for an Expanded Access Programme

PLOS ONE

Dear Dr. Bourner,

Thank you for submitting your manuscript to PLOS ONE. After careful consideration, we feel that it has merit but does not fully meet PLOS ONE’s publication criteria as it currently stands. Therefore, we invite you to submit a revised version of the manuscript that addresses the points raised during the review process.

We look forward to receiving your revised manuscript.

Kind regards,

Om Prakash Choudhary, Ph.D.

Academic Editor

PLOS ONE

Journal Requirements:

2. We note that you have referenced "Institut Pasteur de Bangui. Unpublished data from the monkeypox reference laboratory at Institut Pasteur de Bangui. 2022." which has currently not yet been accepted for publication. Please remove this from your References and amend this to state in the body of your manuscript: (ie “Bewick et al. [Unpublished]”) as detailed online in our guide for authors

"The funders had and will not have a role in study design, data collection and analysis, decision to publish, or preparation of the manuscript."

5. vPlease include your full ethics statement in the ‘Methods’ section of your manuscript file. In your statement, please include the full name of the IRB or ethics committee who approved or waived your study, as well as whether or not you obtained informed written or verbal consent. If consent was waived for your study, please include this information in your statement as well

7. Please ensure that you refer to Figure 1 in your text as, if accepted, production will need this reference to link the reader to the figure.

8. We note you have included a table to which you do not refer in the text of your manuscript. Please ensure that you refer to Tables 1 and 2 in your text; if accepted, production will need this reference to link the reader to the Table.

9 . Please review your reference list to ensure that it is complete and correct. If you have cited papers that have been retracted, please include the rationale for doing so in the manuscript text, or remove these references and replace them with relevant current references. Any changes to the reference list should be mentioned in the rebuttal letter that accompanies your revised manuscript. If you need to cite a retracted article, indicate the article’s retracted status in the References list and also include a citation and full reference for the retraction notice.

Reviewers' comments:

Reviewer's Responses to Questions

**Comments to the Author**

1. Does the manuscript provide a valid rationale for the proposed study, with clearly identified and justified research questions?

Reviewer #1: Yes

2. Is the protocol technically sound and planned in a manner that will lead to a meaningful outcome and allow testing the stated hypotheses?

Reviewer #1: Yes

3. Is the methodology feasible and described in sufficient detail to allow the work to be replicable?

Reviewer #1: No

4. Have the authors described where all data underlying the findings will be made available when the study is complete?

Reviewer #1: Yes

5. Is the manuscript presented in an intelligible fashion and written in standard English?

Reviewer #1: Yes

6. Review Comments to the Author

You may also provide optional suggestions and comments to authors that they might find helpful in planning their study.

Reviewer #1: In this study protocol, 100 patients in the Central African Republic with confirmed monkeypox will be provided tecovirimat and data on clinical signs and symptoms will be collected at baseline, days 4, 8, 14 and 28. Patient outcomes will be assessed on days 14 and 28. Adverse event and serious adverse event data will be summarized. A goal of the study is to gather research information on which to build robust randomized controlled trials.

Minor revisions:

1- Page 19: “arel” is a spelling error.

2- Indicate if adverse event reporting will be conducted according to a standardized method.

3- The statistical analysis plan should be specific. List the statistical methods that will used for summarizing the descriptive analyses, i.e., means, standard deviations, median, first and third quartiles, frequencies and percentages. Be sure to also state how the adverse events data will be summarized. Indicate how variables that are repeatedly measured will be summarized. Will any inferential statistical testing be conducted to compare changes in outcomes?

4- To assist in the review process, add line numbering to the document.

7. PLOS authors have the option to publish the peer review history of their article (what does this mean?). If published, this will include your full peer review and any attached files.

Reviewer #1: No

---

## [Author Response · Author response to Decision Letter 0]

17 Feb 2023

We have provided a response to reviewer letter in the "attach files" section.

---

## [Decision Letter · Decision Letter 1]

27 Mar 2023

Expanded Access Programme for the use of tecovirimat for the treatment of monkeypox infection: a study protocol for an Expanded Access Programme

PONE-D-22-32294R1

Dear Dr. Bourner,

We’re pleased to inform you that your manuscript has been judged scientifically suitable for publication and will be formally accepted for publication once it meets all outstanding technical requirements.

Kind regards,

Sebastien Kenmoe

Academic Editor

PLOS ONE

Additional Editor Comments:Authors have addressed all the questions raised by the reviewers regarding their article. I have carefully assessed the manuscript and response to Reviewers. I believe that the article is now ready for publication.

Reviewers' comments:

Reviewer's Responses to Questions

**Comments to the Author**

1. Does the manuscript provide a valid rationale for the proposed study, with clearly identified and justified research questions?

Reviewer #1: Yes

2. Is the protocol technically sound and planned in a manner that will lead to a meaningful outcome and allow testing the stated hypotheses?

Reviewer #1: Yes

3. Is the methodology feasible and described in sufficient detail to allow the work to be replicable?

Reviewer #1: Yes

4. Have the authors described where all data underlying the findings will be made available when the study is complete?

Reviewer #1: No

5. Is the manuscript presented in an intelligible fashion and written in standard English?

Reviewer #1: Yes

6. Review Comments to the Author

You may also provide optional suggestions and comments to authors that they might find helpful in planning their study.

Reviewer #1: All comments have been adequately addressed.

7. PLOS authors have the option to publish the peer review history of their article (what does this mean?). If published, this will include your full peer review and any attached files.

Reviewer #1: No

---

## [Editor Report · Acceptance letter]

30 Mar 2023

PONE-D-22-32294R1 

Expanded Access Programme for the use of tecovirimat for the treatment of monkeypox infection: a study protocol for an Expanded Access Programme 

Dear Dr. Bourner:

I'm pleased to inform you that your manuscript has been deemed suitable for publication in PLOS ONE. Congratulations! Your manuscript is now with our production department. 

Kind regards, 

on behalf of

Dr. Sebastien Kenmoe 

Academic Editor

PLOS ONE